# Analysis of Bullying in Physical Education: Descriptive Study of Spanish Adolescents

**DOI:** 10.3390/bs14070555

**Published:** 2024-06-30

**Authors:** Juan de Dios Benítez-Sillero, Diego Corredor-Corredor, Álvaro Morente-Montero, Javier Murillo-Moraño, José Manuel Armada-Crespo

**Affiliations:** 1Department of Specific Didactics, Universidad de Córdoba, 14071 Córdoba, Spain; eo1besij@uco.es (J.d.D.B.-S.); eo1momoa@uco.es (Á.M.-M.); m62arcrj@uco.es (J.M.A.-C.); 2Research Group in Sport and Physical Education for Personal and Social Development, 14071 Córdoba, Spain; 3Counselling of Education, Junta de Andalucía, 14071 Córdoba, Spain; dcorcor015@g.educaand.es; 4Teacher Training College “Sagrado Corazón”, University of Córdoba, 14006 Córdoba, Spain

**Keywords:** bullying, observation, physical education, victimisation, perpetration, adolescents

## Abstract

Physical education classes can be a place where both bullying and harassment take place, and a powerful strategy is needed to prevent it. The present study analyses bullying and students’ behaviour as active or passive observers in a general educational context and physical education lessons. A sample of 958 adolescents aged 12 to 18 was studied. A questionnaire was used to analyse the victimisation and the bullying and behaviour observed. The results showed a lower incidence rate than that observed in other studies in physical education classes with a higher level of perpetration by boys than by girls and a more active rejection of aggression by victims and girls. On the other hand, passive attitudes were greater among perpetrators and boys. The observers’ attitudes were similar in both contexts. The scenario in which this subject takes place could reduce the risk of these phenomena occurring. Therefore, physical education teachers should analyse these behaviours and intervene, especially when creating awareness in boys.

## 1. Introduction

Bullying is a phenomenon in which interpersonal aggression among students occurs repeatedly and is sustained over time with an imbalance of power between the aggressor(s) or perpetrators and the victim [1]. It is not a fight or a conflict although this can be a trigger; it is a behaviour or attitude of social exclusion which may involve psychological or verbal abuse in addition to intentional and repeated physical attacks by a schoolchild. The prevalence of bullying is between 10 and 35% [2]. Various factors can influence this variability, such as the behaviours studied and the contexts in which descriptive studies are carried out [3,4]. Bullying is considered one of the most damaging phenomena to students’ well-being at school. Since the first such research was conducted in the 1970s, it has been known that the incidence rates are the highest in the last years of primary school and the first years of compulsory secondary school [5]. The most recent systematic reviews and meta-analyses have indicated that the highest incidence takes place between 11 and 14 years of age [4]. It is also known that there is a gender difference in the profiles of the participants in the phenomenon, with males being more involved in any of the roles, particularly in the role of the aggressor [4,6], but they are equally involved as the victims of their peers.

Bullying has terrible consequences, especially for the victims, such as lower academic performance [4], sleep disorders [4], and even suicidal ideation [7]. However, not only does it affect the victims: it also creates an atmosphere of detachment, coldness, and lack of respect that contaminates school life [8]. This creates a serious problem within the educational community [9]. All scenarios and all activities and communications are affected when, in a school, negative interpersonal relationships and socio-moral issues, which are implicit in the phenomenon of school bullying, are allowed to continue [10,11], but some scenarios and some activities, due to important factors such as communication and interactive relationships, seem to be more exposed to the processes of bullying and harassment.

However, when a physical education (PE) class is carried out in a cordial environment and aspects of interpersonal relationships are considered relevant educational factors [12] in an appropriate way, the activities and the social interactions that they imply seem to contribute to a better coexistence climate and a lower incidence of bullying [13,14]. On another note, there is some evidence that students excluded from PE classes due to health problems suffer more bullying at school and are even bullied more than those who participate normally in their school PE classes [15]. Some studies have indicated that the prevalence of bullying in PE classrooms may be high if the class is not properly managed; specific scenarios (playgrounds, gymnasiums, and external exits) seem to have less teacher vigilance toward students, who often interact in large or closed spaces (locker rooms) where teachers do not enter [16,17,18].

Some research has focused on the study of bullying and PE, physical activity, and sports in adolescents [17,19] with controversial results. Studies such as [20,21,22], among others, have used disparate methodologies and procedures, producing equally mixed results. For example, questions can be asked about the occurrence of bullying over an entire school year or a limited situation of two months; the victims or perpetrators are considered if the occurrence is only on a one-off occasion (which is excluded from the definition of bullying itself) or if it is something that occurs with a certain frequency and involves the same protagonists (aggressor(s) and victim(s)). In just one study [21], the occurrence of bullying was analysed both at a general level and in PE classes for the same sample, and it obtained results where, compared to the 23% perpetrators and 39% victims at the general level in the whole-school context, in the PE classes, 15% were perpetrators and 28% were victims among the same students.

Bullying is considered a group phenomenon which, in addition to bullies and victims, involves many participants who are, to some extent, informed or directly witness the events and are more or less inhibited or involved [23]. In this regard, it has been described that observers can adopt the following four roles: (1) assistants, who join the side of the bully; (2) reinforcers, who encourage the bully and reinforce the bullying behaviour with their support; (3) passive observers, who simply watch what is happening; and (4) advocates or proactive observers that are against what is happening and stand up on behalf of the victims to try to prevent a repetition of the events, explicitly supporting the victim [23]. When peers intervene in situations of bullying (i.e., exclusion or unjustified aggression) by reproaching the bully, expressing such behaviour is not permissible, or alerting teachers to intervene, these bullying behaviours decrease significantly, although this proactive attitude of an observer is not common in this complex phenomenon [24]. Moreover, the profiles and attitudes of observers are less studied than anything related to the profiles and attitudes of the protagonists. More is known about victimisation and perpetration than about what affects the observers of the phenomenon in general [25]. Similarly, there is more knowledge about passive bystanders or observers and proactive defenders as categories within bullying, differentiating them from the other roles, but few studies have analysed the observational attitudes of all those involved [26,27]. We also must consider that victims or perpetrators may be victims or perpetrators in specific situations at school, but they may also act as protagonists or in another role in the social fabric of their school groups. Being an observer of bullying situations can even have negative personal and social consequences for the individual due to the vicarious impacts [28,29,30].

This study aimed to analyse the occurrence of bullying in PE classes compared to general bullying in schools, focusing on analysing the behaviours and roles of observers in both educational contexts. It is hypothesised that PE is an educational setting that, being not free from the risks of bullying, is less likely to present a lower incidence than bullying in general at school, with perpetration being higher among boys. It is also presumed that observers’ profiles and behaviours are more active among victims and girls.

## 2. Materials and Methods

### 2.1. Design and Participants

A descriptive, cross-sectional study design with non-probability sampling was carried out. A total of 958 students aged between 12 and 18 (50% girls M = 14.65, SD = 1.36) participated in the study. Participants were selected from four public schools in Huelva and Córdoba (Spain) due to accessibility for the researchers. The participants ranged from the first to the fourth year of compulsory secondary education (12–16 years old) and the first year of upper secondary education (17–18 years old). No exclusion criteria were applied, accepting all collected questionnaires. The study was carried out in the academic year 2020–2021, during March to May 2021. In this year, the restrictions due to the COVID-19 pandemic consisted of the use of masks, the predominance of the use of outdoor spaces, the limitation of the use of changing rooms in shifts, and increased hygienic measures.

The study was conducted as a part of an educational research project funded and authorised by the Regional Ministry of Education and Sport of the Andalusian Regional Government. All the national and international ethical standards such as the Declaration of Helsinki and personal data protection laws were followed. Permissions to conduct the research project were granted by the school boards, and informed consents were provided by the families of the participants. The objective of the study was explained to the participants, and the anonymous, confidential, and voluntary nature of their participation was stressed. The project was approved by the Ethics Committee of the University of Córdoba.

### 2.2. Instruments

#### 2.2.1. Bullying

The Spanish version of the European Bullying Intervention Project Questionnaire (EBIPQ) [1] was used to measure bullying victimisation and perpetration at a general level. It includes 14 items of which seven are focused on victimisation and seven are focused on perpetration. Questions were answered on a Likert-type response scale that ranges from 0 to 4, (0 = never, 1 = once or twice, 2 = once or twice a month, 3 = about once a week and 4 = more than once a week). This questionnaire obtained good Cronbach’s alphas for victimisation (α = 0.84) and perpetration (α = 0.82). Students were considered to be perpetrators or victims if they claimed to have answered once or twice a month or more frequently for any of the behaviours presented for victimisation or perpetration. Students who were involved in both victimisation and perpetration at that frequency were considered perpetrator–victimised. This data collection methodology was used because it was validated with good psychometric properties and highly cited in peer-reviewed journals [2,4,8,13].

The same questionnaire was used, but with the nuance “in PE classes”, to investigate the same phenomena (victimisation/perpetration) occurring in PE lessons. To date, this study represents the first attempt to analyse bullying in physical education using this questionnaire; thus, there is no known evidence regarding its utilisation in this context. Therefore, it was necessary to conduct a confirmatory factor analysis of the current data, which showed they adjusted properly: $\chi_{91}^{2}$ = 10,938.15, *p* = 0.022; χ^2^/gl = 120.20; NNFI = 0.997; CFI = 0.998; TLI = 0.997; RMSEA = 0.019, 90% CI [Li = 0.008, Ls = 0.028], *p* = 1.000 y SRMR = 0.074. The Cronbach’s alpha tests for victimisation (α = 0.85) and perpetration (α = 0.76) showed good internal consistency. To ensure that students did not confuse harassment experiences in PE with those in general school environments, the questionnaire assessing bullying in PE was administered after the questionnaire assessing harassment in general environments. This sequence allowed students to notice that it was the same question but with a different purpose. Additionally, it was indicated in bold and large font that the questions specifically referred to PE. The researcher administering the questionnaires also verbally emphasised that, after responding in the general educational context, they should exclusively respond about bullying behaviours in the PE classes.

#### 2.2.2. Observation Bullying

To analyse the role of observers in bullying situations, the two dimensions proposed by Caballo et al. [26] in the following instrument “Multimodal School Interaction Questionnaire” (CMIE-IV) were used. This questionnaire consists of five factors, although only two of them were considered. The first of the dimensions considered was the active observation in defence of the bullied factor, in which 6 items were taken into account. The second factor was the passive observation in which 4 items were considered. In this questionnaire, the subjects were asked “How many times have you experienced these situations in the last two months?” The response options were (1 = never, 2 = few, 3 = quite a few and 4 = many). As with the bullying questionnaire, subjects were first asked about the occurrence of these behaviours at a general level. A confirmatory factor analysis of the current data showed they adjusted properly: $\chi_{45}^{2}$ = 24,846.97, *p* < 0.001; χ^2^/gl = 512.15; NNFI = 0.997; CFI = 0.997; TLI = 0.997; RMSEA = 0.045, 90% CI [Li = 0.035, Ls = 0.055]; *p* = 0.782; and SRMR = 0.047. The Cronbach´s alpha tests for active observation in defence of the harassed (α = 0.88) and passive observation (α = 0.79) showed good internal consistency. Subsequently, the same questionnaire was carried out at the level of the PE class. A confirmatory factor analysis of the current data showed they adjusted properly: $\chi_{45}^{2}$ = 67,717.87, *p* < 0.001; χ^2^/gl = 1504.84; NNFI = 0.999; CFI = 0.999; TLI = 0.999; RMSEA = 0.037, 90% CI [Li = 0.026, Ls = 0.048], *p* = 0.782 y SRMR = 0.040. The Cronbach´s alphas tests for active observation in defence of the harassed (α = 0.93) and passive observation (α = 0.86) showed good internal consistency.

### 2.3. Statistical Analysis

First, after checking the non-normality of the sample data, Spearman’s bivariate correlations of the quantitative variables were analysed. Student’s t-test to contrast the existence of differences by gender, and Cohen’s d was considered to control the effect size. To determine the relationship between observation as a function of the roles of involvement in bullying, an ANOVA analysis with Tukey’s post hoc test was carried out. Subsequently, a linear regression analysis was performed with the dependent variable being observation and the independent variables being sex, victimisation and bullying perpetration as quantitative variables. The analyses were carried out using the IBM SPSS Statistics 25 statistical package.

## 3. Results

Table 1 shows the bivariate correlations among the different study variables. The negative relationships between age or year with victimisation, both in general and in PE, and the decrease in the active role observers stand out.

Table 2 shows the variables of victimisation and perpetration of bullying as a function of gender. Both in the general exploration and in the exploration in the PE classes, perpetration was higher in boys. There were no significant differences in the sample in victimisation.

Table 3 shows the differences in the role of observers according to gender. Overall, girls showed higher levels of active observation and lower levels of passive observation. In PE, only passive observation was significant, being lower in girls.

Table 4 shows the occurrence of bullying at a general level in the whole educational context with the victims being 18% plus 6% of the perpetrators–victimised, totalling 24%, and the perpetrators 2% plus 6% of the perpetrators–victimised, i.e., 8%. The percentage of students not involved was 74%. Subjects are also categorised according to the roles they adopt in bullying at a general level and then compared according to their attitude as observers. In relation to active observation, when comparing according to the roles adopted in bullying, this was significant (F = 12.79; *p* ≤ 0.001), and average differences were found among some groups with higher values for victims compared to those not involved. The passive attitude was also significant (F = 5.37; *p* ≤ 0.001); at the group level, the perpetrators–victimised presented higher values than those not involved at the mean level.

Table 5 shows the occurrence of bullying in PE and compares the different groups according to their attitude as observers in PE classes according to the roles adopted in bullying. Differences were found in the active observation attitude (F = 7.04; *p* ≤ 0.001) with higher levels in the victims than in those not involved. In the passive attitude, there were also significant differences (F = 8.09; *p* ≤ 0.001) with the non-involved showing lower values than the victims and perpetrators.

Table 6 represents the attitude of active and passive observation at the general level as a dependent variable in a linear regression analysis with the independent variables of grade, gender, victimisation, and perpetration. Relationships are found for active observation with the student’s grade, being lower as one moves up to higher grades, and a positive relationship with general victimisation. In relation to passive observation, it was lower in boys and was related to perpetration behaviour. The year of study variable was used instead of age, as it showed higher levels of significance.

Finally, Table 7 presents active and passive observation in PE as the dependent variable in a linear regression analysis with the independent variables grade, gender, victimisation and perpetration in PE. The active observation was negatively related to the grade studied and positively related to victimisation. In passive observation, lower levels were found in girls, and higher levels were found in those who presented greater perpetration behaviour.

## 4. Discussion

In this study, we have focused on the PE class as a particularly relevant setting and the prevention of bullying. Although it is presumed that PE lessons are not free from this phenomenon of mistreatment among students, its incidence is differential both for the protagonists of the problem and for the observers of it. Based on the results obtained, we can argue, with reference to the levels of occurrence that other studies [3] have shown, that the general trend indicating that more than three quarters of the school population is unaffected by the phenomenon of bullying, i.e., they state that they are neither involved in nor affected by these problems, is fulfilled. Nevertheless, this prevalence is lower in the PE classes in this sample: less than one schoolchild out of ten is involved in bullying in any of its protagonist roles (victim/bully). This confirms our initial hypothesis, as in the only previous study [21] that analysed this difference. The trends previously reported by other studies [6,8,20] that indicate that boys are more involved (both in general and PE classes) than girls are also maintained. Indeed, boys showed higher values in perpetration behaviour both at the general and PE level with no significant differences in victimisation according to gender. That is to say, the role of victims is evenly distributed between girls and boys in PE classrooms.

Concerning the occurrence of school bullying at a general level, the data from our study indicate that 26% of adolescents of these ages have been framed within the leading roles involved in it. These data are very similar to studies that place the prevalence at around 25–30% [31]. It is noteworthy that the incidence of bullying seemed to remain constant in recent years [2,32]. In our study, boys were more involved in aggressive behaviour than girls, as in other studies [6,8,33], although there were no significant differences with respect to victimisation as in recent studies in adolescents [8,13]. In relation to the age and school year of the participants, less victimisation was observed as age and school year increased. However, there was no significant relationship between age and school year with perpetration behaviour. These results align with other studies regarding victimisation [8,26], but not with perpetration, where it seems that perpetration behaviour increases with age [26,33].

Regarding the occurrence of bullying in PE classrooms, the overall prevalence was 8% (including 1% of perpetrators and 1% of victimised perpetrators). These results are lower than the few previous studies where bullying in PE classrooms has been analysed. But as we have seen, only one previous study analysed the occurrence of bullying in the same population at the general level and in PE [21]. In this study, 15% of 10–15-year-olds were perpetrators in PE classes compared to 23% overall, a difference of 8 percentage points, while in our study, the difference was 6 points but with lower results both in PE (2%) and overall (8%). In relation to victimisation, the Gano-Overway study [21] reported a prevalence of 28% in PE versus 39% overall, while in our study, the prevalence of both was lower (7% PE and 24% overall). However, it has already been noted that the methodology was different in terms of the time being investigated. The Gano-Overway [21] study asked about the occurrence in the last 30 days and considered that bullying was prevalent when they answered that such behaviour had occurred at least once, while our study considered the perpetrator or victim to be those who indicated that the events had occurred two or more times a month during the last two months. Therefore, the probability of prevalence in our study is clearly lower and more in line with the concept itself that defines bullying [34,35], which requires repetition over a clearly defined period of time. The use of such a cut-off point to consider the involvement in each bullying role in our study is argued by the definition of bullying itself, which requires the occurrence to be repeated and sustained over time, as well as by the greater psychometric value of the scale used [1].

With respect to other studies that have also focused on bullying in PE classes [22], studying Portuguese students aged between ten and eighteen, considering the period of the entire academic year, and recording the event as bullying when there is at least one event, they found that 30% of students reported victimising behaviours and 29% reported having carried out perpetrating behaviours. Therefore, the results were higher in incidence in our study, although this could be explained by the fact that it was asked for the occurrence in a full academic year and in our study in only two months. It was also considered involved if the behaviour is experienced or carried out only once, while in our study, it takes place once or twice a month.

The most recent study analysing bullying in PE is that of Borowiec et al. [20] in Poland with adolescents aged 14–16. In this one, as in ours, they asked about bullying in the last two months, considering the person involved if they answered “rarely” or “often”, categories 2 and 3 of a three-dimensional scale. In this study, in contrast to those previously reviewed, the percentage of perpetrators was 23% higher than that of victims 19%.

The explanations for the lower levels of occurrence of both victimisation and perpetration in physical education in our study, compared to previously published studies [20,21,22], could be due in part to methodological issues and the very concept of the phenomenon of bullying itself, since the other studies asked about a different time period and the repetition of bullying was excluded, which is a defining characteristic of bullying. The instrument used here, the EBIPQ, defines a specific, standardised, and validated period for the measurement of the prevalence of bullying in any setting, and it includes repetition of aggression as a constituent element of the construct, as mentioned above. Other possible explanations could be that this study was carried out when some measures of social distance in PE were still in place due to the COVID-19 pandemic. For example, students did not use the locker rooms to a large extent, and they did so without crowding. Some studies claim that many bullying situations occur in the locker rooms in PE because they are outside the supervision of teachers [16,17,18].

In relation to gender and grade, our study found that there are no significant differences in the victimisation of students in PE classes. However, other studies found, both in this specific scenario and at a general level, that boys recognised themselves as more victimised than girls, coinciding with what occurs in other studies on bullying at a general level [8,36]. In reference to the perpetration in our study, boys were more perpetrators than girls, which is consistent with other studies [20,37].

As far as grade is concerned, our results found greater victimisation as the grade increased, both in physical education classes [20] and at a general level [8]. Differences in perpetration were not significant with grade, contrasting with other studies in PE that claim that violence and perpetration tendencies increase with grade [20,38].

When we moved on to the analysis of active or passive observation behaviour in bullying situations, girls showed higher levels of proactive observation in general bullying, which were differences that were not found in PE classes. On the other hand, in passive observation, boys showed higher values both in general level and in PE. This coincides with the results of other studies that analyse it at a general level in the whole educational context [26,27,39] with no analysis in physical education.

With grade, proactive observation decreases both in general and in PE classrooms, which is consistent with what has been stated in other studies [26,27]. In relation to passive observation, we found no significant differences as published in the study by Parris et al. [27], in contrast to Caballo et al. [26], in which passive observation increased with grade. When studying the relationships among the attitude as observers depending on the roles in bullying, higher levels of proactive observation were found in victims, which agree with the results of Wu et al. [30]. We interpret these results, which reinforce previous studies, in a double sense. On the one hand, perhaps victims have, due to their experience, more pronounced peer-supportive behaviours compared to non-involved and perpetrators [30]. On the other hand, perhaps some of these victims belong to that group that has been given different names (paradoxical victims, victimised aggressors) that participate in traits of both roles. A complementary explanation would suggest that perhaps some victims have a higher level of empathy and anti-bullying sentiment [40,41]. In any case, victims’ experiences make them more familiar with the signs of bullying, and they may be better able to detect it in order to act, than perpetrators or those not involved [42].

In relation to passive observation, it was higher in boys and among students with more aggressive behaviour, which was also replicated for PE victimisation. This passive attitude of perpetrators when observing other bullying situations in which they are not involved could again be associated with the perpetrators’ lack of empathy [40], which is consolidated, uncontroversial knowledge among previous research. In our study, the highest levels of passive at a general level were found in victimised perpetrators. Nonetheless, in physical education, differences were significant also between victims and non-victims and were related to victimisation. In the absence of previous studies in this area, it is risky to seek an explanation, and more knowledge and research are needed. Stein and Jimerson [11] found that perpetrators had the highest levels of moral disengagement in bullying situations, which could also be related to this passive attitude [40]. Another aspect that could explain these relationships could be the lower social competences and skills of bystanders who display these passive attitudes. Likewise, this attitude that leads them to ignore bullying situations could be due to a lower knowledge or ability to intervene in defence of the victim [39]. For this reason, teachers should not assume that young people who do not intervene in defence of others do so not because they are unwilling to do so but perhaps because they are unable to find the mechanisms to intervene. It is therefore important to know whether young people know how to intervene in a bullying situation when they are bystanders and how they feel about it [39]. It is also desirable that in-service teachers and future teachers receive training on bullying in physical education [43].

## 5. Conclusions

From the results, and in relation to the objectives that were set out, this paper can point to several conclusions: bullying in PE classes in our study was lower than the prevalence levels that have been published by the few that exist in the scientific literature that have focused on the specific scenario of this area of the school curriculum. There are methodological and construct differences that may be at the basis of this difference in prevalence, as we have mentioned. Yet, we also believe that the scenario itself, which is more open to physical activity, and perhaps for many students is more fun and relaxing (although obviously not for all), reduces the risk of the emergence of these phenomena. Therefore, we will have to continue studying this incidence in the future to find the keys to this decrease in risk and learn to transfer these keys to other educational scenarios.

Boys, as all studies admit, also showed higher levels of perpetration than girls in PE, but this was not replicated among girls in terms of victimisation. Active observation behaviour was higher among victims of bullying and among girls. Passive observing behaviour was higher among perpetrators and boys. The dynamics of observers’ attitudes were similar in the general educational context and in PE classes. Therefore, PE teachers should analyse and intervene especially in raising the awareness of boys.

With this study, we have tried to provide a more in-depth understanding of what bullying in PE classes is like in comparison with that which occurs at a general level and the attitude of observation of all those involved. But more studies are needed to know the specificity of this phenomenon, since the subject of PE, due to its specific characteristics, can become both a space where bullying can occur and a powerful tool to promote strategies and values in students that help against this problem.

This study is limited by the specificity of the sample, which has not been randomised. Questionnaire-based measurement of the occurrence of bullying may not be as accurate, so conclusions should be taken with caution. Nevertheless, the study presented here shows evidence of the importance of the settings in which educational activities take place in the school context. In this sense, there is no doubt that the setting of PE is particularly interesting as a framework in which, even with a certain level of risk, bullying can be reduced. This will increase if, as Jiménez-Barbero et al. [17] state, PE programmes continue to include social competence and moral values among their objectives and intentions. Furthermore, other practical measures to reduce bullying in the physical education classroom could involve implementing supervision strategies in the locker rooms through designated student monitors, as well as adopting a teaching style that is less controlling [41] and more open to activities with adaptable difficulty levels, allowing students to feel comfortable in class.

Finally, it is important to consider the possible political implications derived from the findings. As noted, the subject of physical education provides an opportunity to implement school-based anti-bullying programs, which could be crucial in preventing and reducing bullying in physical education as well as in other educational settings. Therefore, it would be necessary to direct new modifications in educational laws where sufficient importance is given to this subject to achieve not only the physical benefits it provides but also social well-being. Additionally, the adoption of specific measures in school policies that reflect the commitment of schools as safe environments where tolerance and respect are promoted is essential. This underscores the need to train current and future teachers not only in physical education but in all areas so they can design joint intervention programs with the aim of preventing and reducing school bullying. It is therefore crucial that these measures are not limited to reactive action in bullying situations but also focused on active prevention.

## Figures and Tables

**Table 1 behavsci-14-00555-t001:** Spearman’s correlations between all studied variables.

Variable	1	2	3	4	5	6	7	8	9
1. Age	-	-	-	-	-	-	-	-	-
2. Grade	0.915 **	-	-	-	-	-	-	-	-
3. Victimisation GE	−0.084 **	−0.132 **	-	-	-	-	-	-	-
4. Perpetration GE	0.008	−0.034	0.531 **	-	-	-	-	-	-
5. Victimisation PE	−0.070 *	−0.103 **	0.559 **	0.399 **	-	-	-	-	-
6. Perpetration PE	−0.029	−0.044	0.321 **	0.567 **	0.436 **	-	-	-	-
7. Acti. Observ. GE	−0.115 **	−0.144 **	0.214 **	0.066 *	0.106 **	0.020	-	-	-
8. Pasi. Obersv. GE	0.006	−0.010	0.127 **	0.219 **	0.104 **	0.166 **	0.195 **	-	-
9. Acti. Observ. PE	−0.142 **	−0.164 **	0.169 **	0.057	0.108 **	0.036	0.764 **	0.210 **	-
10. Pasi. Obersv. PE	−0.013	−0.052	0.172 **	0.251 **	0.154 **	0.184 **	0.179 **	0.717 **	0.373 **

Notes. GE = general; PE = physical education; Acti. = active; Pasi. = passive; Observ. = observation. * Established level of significance: *p* < 0.05; ** Established level of significance; *p* < 0.01.

**Table 2 behavsci-14-00555-t002:** Descriptive statistics and differences by gender for all variables of bullying.

Variable/Condition	Total (n = 958)		Boys (n = 479)		Girls (n = 479)				
	M	SD	M	SD	M	SD	*t*	*p*	*d*
Victimisation general	0.40	0.59	0.38	0.61	0.41	0.58	−0.55	0.581	−0.05
Perpetration general	0.16	0.35	0.19	0.40	0.13	0.30	20.83	0.005 **	0.17
Victimisation PE	0.12	0.35	0.13	0.38	0.12	0.82	0.28	0.780	0.02
Perpetration PE	0.05	0.18	0.07	0.22	0.04	0.13	20.08	0.038 *	0.39

Notes. M = mean; SD = standard deviation; *t* = Student’s t; *d* = Cohen’s d. * Established level of significance: *p* < 0.05; ** Established level of significance; *p* < 0.01.

**Table 3 behavsci-14-00555-t003:** Descriptive statistics and differences by gender for all variables of observation.

Variable/Condition	Total (n = 958)		Boys (n = 479)		Girls (n = 479)				
	M	SD	M	SD	M	SD	*t*	*p*	*d*
Active observation general	2.63	0.83	2.58	0.81	2.69	0.85	−2.05	0.041 *	−0.13
Passive observation general	1.83	0.59	1.93	0.62	1.74	0.55	4.99	≤0.001 ***	0.32
Active observation PE	2.50	0.95	2.47	0.92	2.53	0.98	−0.93	0.353	0.06
Passive observation PE	1.78	0.65	1.88	0.69	1.67	0.58	5.08	≤0.001 ***	0.33

Notes. M = mean; SD = standard deviation; t = Student’s t; d = Cohen’s d; PE = physical education. * Established level of significance: *p* < 0.05; *** Established level of significance: *p* ≤ 0.001.

**Table 4 behavsci-14-00555-t004:** Comparisons of their observational attitude in terms of their role in general bullying.

Condition		Active Observation General		Passive Observation General	
	N (%)	M (SD)	Post-Hoc	M (SD)	Post-Hoc
0. Not involved	709 (74)	2.55 (0.85)	1 **	1.80 (0.59)	3 *
1. Victims	172 (18)	2.97 (0.68)	0 **	1.91 (0.55)	
2. Perpetrators	20 (2)	2.44 (0.99)		1.97 (0.72)	
3. Perpetrators-Victims	57 (6)	2.77 (0.74)		2.07 (0.62)	0 *

Notes. M = mean; SD = standard deviation. * Established level of significance: *p* < 0.05; ** Established level of significance; *p* < 0.01.

**Table 5 behavsci-14-00555-t005:** Comparisons of their observational attitude according to their role in bullying in PE.

Variable/Condition		Active Observation PE		Passive Observation PE	
	N (%)	M (SD)	Post-Hoc	M (SD)	Post-Hoc
0. Not involved	880 (92)	2.45 (0.96)	1 ***	1.74 (0.64)	1 ** 0.2 *
1. Victims	53 (6)	3.02 (0.62)	0 ***	2.06 (0.63)	0 **
2. Perpetrators	13 (1)	2.88 (0.79)		2.30 (0.77)	0 *
3. Perpetrators-Victims	12 (1)	2.71 (0.70)		2.12 (0.71)	

Notes. M = mean; SD = standard deviation. * Established level of significance: *p* < 0.05; ** Established level of significance: *p* < 0.01; *** Established level of significance: *p* ≤ 0.001.

**Table 6 behavsci-14-00555-t006:** Linear regression with grade, gender, victimisation, and perpetration as predictors of behaviour when observing bullying situations in general.

Variable/Condition	Active Observation General		Passive Observation General	
	β	t	β	t
Grade	−0.077 ***	−3.656	0.02	0.108
Gender (female)	0.093	1.772	−0.174 ***	−4.610
Victimisation general	0.292 ***	5.516	0.021	0.562
Perpetration general	−0.136	−1.545	0.235 ***	3.716
	F = 14.51 *p* < 0.001 R^2^ = 0.06		F = 12.16 *p* < 0.001 R^2^ = 0.05	

Notes. *** Established level of significance: *p* ≤ 0.001.

**Table 7 behavsci-14-00555-t007:** Linear regression with grade, gender, victimisation and perpetration as predictors of behaviour when observing bullying situations in PE.

Variable/Condition	Active Observation PE		Passive Observation General	
	β	t	β	t
Grade	−0.117 ***	−4.824	−0.015	−0.917
Gender (female)	0.056	0.927	−0.200 ***	−4.881
Victimisation PE	0.285 *	2.934	0.144 *	−2.166
Perpetration PE	−0.007	−0.069	0.421 **	−3.252
	F = 9.36 *p* < 0.001 R^2^ = 0.04		F = 13.93 *p* < 0.001 R^2^ = 0.06	

Notes. PE = physical education. * Established level of significance: *p* < 0.05; ** Established level of significance: *p* < 0.01; *** Established level of significance: *p* ≤ 0.001.

## Data Availability

The data presented in this study are available on request from the corresponding author. The data are not publicly available due to ethical and privacy considerations to protect the confidentiality of participants.

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
