# Peer review of "Analysis of Bullying in Physical Education: Descriptive Study of Spanish Adolescents"

_behavsci, 2024, doi:10.3390/bs14070555_

Round 1
Reviewer 1 Report
Comments and Suggestions for Authors
This paper provides a compelling study on bullying and observer behaviors in Physical Education (PE) classes, revealing a lower incidence of bullying in PE compared to general education settings. It has the hallmarks of a well-executed research study and, following some revisions, will be suitable for publication in Behavioral Sciences.
Materials and Methods
Participants: The paper presents an adequate sample size. However, providing more detail on recruitment methods and participant selection criteria are needed.
Instruments: Use of validated questionnaires adds reliability. However, providing more evidence about the application of these instruments in the school and PE context and with the target population in previous research might enhance the study’s relevance and accuracy.
The study uses the same instrument to measure bullying in both general school settings and PE, with the primary distinction being the emphasis on PE within the prompt. A key concern is ensuring students do not conflate bullying experiences in PE with those in general school settings. Authors should clearly explain how they distinguish between the two contexts to ensure accurate and context-specific responses from the students.
Conclusion
While the practical implications are well-articulated, additional specific recommendations for how teachers (e.g, PE and classroom teacher) can implement strategies to reduce and prevent bullying would be beneficial. Particularly, the lower incidence of bullying in PE classes was partly attributed to the reduced use of locker rooms. Suggestions for PE in a non-COVID setting are recommended.
Including a discussion on potential policy implications would indeed enrich the conclusion. This could involve advocating for the implementation and promotion of school-wide anti-bullying policies and programs. Additionally, addressing policies on inclusion and support for marginalized students could be crucial in fostering a safe and supportive learning environment for all.
Reviewer 2 Report
Comments and Suggestions for Authors
Brief summary:
Few articles examine the prevalence of bullying in physical education classes. So your article is very important in that sense. The distinction between active and passive bystanders is very interesting. The use of validated tools is a real addition to your article.
Article:
A third hypothesis could be added, namely to compare differences by gender.
I also wonder why the Bonferroni correction was not carried out.
For the results section, I suggest the following order : prevalence, spearman, t test, ANOVA
Review:
I suggest this reference : Wei, M., & Graber, K. C. (2023). Bullying and Physical Education : A Scoping Review. Kinesiology Review, 1‑18. https://doi.org/10.1123/kr.2022-0031
In many places in the article, references could be added.
Specific comments:
L27-29 : I suggest that you clearly define bullying and to include psychological/verbal abuse.
L36-38: references are missing
L50 : a paragrah on personal consequences of bullying could be added
Line 61: there is an increasing number of scientific papers on violence in sport. It could be useful
L98 : (...) it is less (...)
L99 : H2 could be cleary identified and more precise : observers are more active than ...
L34 : reference is missing. You could add more details on psychometric properties
L269-L272: reference is missing
L274: I suggest you specify that the results apply to the sample and not to the population.
L276-278: reference
L289: I suggest that you replace subjects by participants
L288-292 : There are three ideas in the same sentence. I suggest you cut it into three.
L304: replace prevalence by prevalent
L351: I propose that H2 be clearly identified
Comments on the Quality of English Language
Suggestions:
Replace schoolchildren by students
Replace victimisation by victimization
Replace observers by bystanders
